# HLA-Matched Allogeneic iPS Cells-Derived RPE Transplantation for Macular Degeneration

**DOI:** 10.3390/jcm9072217

**Published:** 2020-07-13

**Authors:** Sunao Sugita, Michiko Mandai, Yasuhiko Hirami, Seiji Takagi, Tadao Maeda, Masashi Fujihara, Mitsuhiro Matsuzaki, Midori Yamamoto, Kyoko Iseki, Naoko Hayashi, Ayumi Hono, Shoko Fujino, Naoshi Koide, Noriko Sakai, Yumiko Shibata, Motoki Terada, Mitsuhiro Nishida, Hiromi Dohi, Masaki Nomura, Naoki Amano, Hirokazu Sakaguchi, Chikako Hara, Kazuichi Maruyama, Takashi Daimon, Masataka Igeta, Toshihiko Oda, Utako Shirono, Misato Tozaki, Kota Totani, Satoshi Sugiyama, Kohji Nishida, Yasuo Kurimoto, Masayo Takahashi

**Affiliations:** 1Department of Ophthalmology and Kobe City Eye Hospital, Kobe 650-0047, Japan; sunao.sugita@riken.jp (S.S.); michiko.mandai@riken.jp (M.M.); yhirami@kcho.jp (Y.H.); tigerseiji@gmail.com (S.T.); tadao_maeda@kcho.jp (T.M.); masashifujihara@hotmail.co.jp (M.F.); mmatsuzaki@me.com (M.M.); midori_yamamoto@kcho.jp (M.Y.); naoshi.koide@riken.jp (N.K.); ykurimoto@mac.com (Y.K.); 2Department of Ophthalmology, Kobe City Medical Center General Hospital Kobe 650-0047, Japan; 3Laboratory for Retinal Regeneration, RIKEN Center for Biosystems Dynamics Research, Kobe 650-0047, Japan; kyoko.iseki@riken.jp (K.I.); naoko.sp@gmail.com (N.H.); ayumi.hono@riken.jp (A.H.); shoko.fujino@riken.jp (S.F.); noriko.sakai@riken.jp (N.S.); yumiko.shibata@riken.jp (Y.S.); motoki.terada@riken.jp (M.T.); mituhiro.nishida@riken.jp (M.N.); 4Department of Ophthalmology, School of Medicine, Toho University, Tokyo 153-8515, Japan; 5Center for iPS Cell Research and Application, Kyoto University, Kyoto 606-8567, Japan; hiromi.dohi@cira.kyoto-u.ac.jp (H.D.); mnomura@cira.kyoto-u.ac.jp (M.N.); n.amano@cira.kyoto-u.ac.jp (N.A.); 6Department of Advanced Device Medicine, Graduate School of Medicine, Osaka University, Osaka 565-0871, Japan; sakaguh@ophthal.med.osaka-u.ac.jp (H.S.); chikako.ueno@ophthal.med.osaka-u.ac.jp (C.H.); 7Department of Innovative Visual Science, Graduate School of Medicine, Osaka University, Osaka 565-0871, Japan; kazuichi.maruyama@ophthal.med.osaka-u.ac.jp; 8Department of Biostatistics, Hyogo College of Medicine, Nishinomiya 663-8501, Japan; daimon@hyo-med.ac.jp (T.D.); ma-igeta@hyo-med.ac.jp (M.I.); 9Center for Clinical Research and Innovation, Kobe City Medical Center General Hospital, Kobe 650-0047, Japan; t-oda@kcho.jp (T.O.); ushirono@kcho.jp (U.S.); m.tozaki.kcho@gmail.com (M.T.); 10Tomey Corporation, Nagoya 451-0051, Japan; kt-totani@tomey.co.jp (K.T.); s-sugiyama@tomey.co.jp (S.S.); 11Department of Ophthalmology, Graduate School of Medicine, Osaka University, Osaka 565-0871, Japan; knishida@ophthal.med.osaka-u.ac.jp

**Keywords:** HLA, immune reactions, macular degeneration, iPS cells, retina, transplantation

## Abstract

Immune attacks are key issues for cell transplantation. To assess the safety and the immune reactions after iPS cells-derived retinal pigment epithelium (iPS-RPE) transplantation, we transplanted HLA homozygote iPS-RPE cells established at an iPS bank in HLA-matched patients with exudative age-related macular degeneration. In addition, local steroids without immunosuppressive medications were administered. We monitored immune rejections by routine ocular examinations as well as by lymphocytes-graft cells immune reaction (LGIR) tests using graft RPE and the patient’s blood cells. In all five of the cases that underwent iPS-RPE transplantation, the presence of graft cells was indicated by clumps or an area of increased pigmentation at 6 months, which became stable with no further abnormal growth in the graft during the 1-year observation period. Adverse events observed included corneal erosion, epiretinal membrane, retinal edema due to epiretinal membrane, elevated intraocular pressure, endophthalmitis, and mild immune rejection in the eye. In the one case exhibiting positive LGIR tests along with a slight fluid recurrence, we administrated local steroid therapy that subsequently resolved the suspected immune attacks. Although the cell delivery strategy must be further optimized, the present results suggest that it is possible to achieve stable survival and safety of iPS-RPE cell transplantation for a year.

## 1. Introduction

Immune cells are capable of discriminating between self and non-self cells by recognizing major histocompatibility complex (MHC) antigens that are expressed on the cell surface. In humans, these cell surface antigens indicate human MHC (=human leukocyte antigens (HLA)). When immune cells encounter MHC molecules they do not recognize, an immune response is triggered in order to get to rid the body of the foreign entity. Although this mechanism is important for protecting against antigens from pathogens, it can be a hindrance when performing cell transplantations from one individual to another. Recently, we experimentally demonstrated that immune responses could be avoided when retinal pigment epithelial (RPE) cells generated from iPS cells derived from an MHC homozygous donor were transplanted to a recipient with matched MHC [1,2,3]. Using RPE cells derived from MHC homozygote monkey iPS cells, we developed an in vivo experimental method that could be used to test the immune-response in primates [1,3]. We also established an in vitro evaluation protocol to test the graft immunogenicity by co-culturing human iPS-RPE graft cells with human immune cells including T cells from the recipients [2]. Results from these preclinical studies showed that MHC (HLA)-class II molecules, as well as class I, were critical for the mechanisms of RPE cells-related immune rejections after transplantation. In fact, RPE cells constitutively express MHC class I, and inducibly express MHC class II during the stress of inflammation (e.g., rejection) [1,2], making these cells potential targets for T lymphocytes.

Globally, age-related macular degeneration (AMD) is the leading cause of blindness in elderly populations [4,5,6,7]. Standard treatments currently used for patients with exudative AMD (wet AMD) include intravitreous injections of anti-vascular endothelial growth factor (VEGF) drugs, photodynamic therapy, or a combination of both. However, anti-VEGF treatments often require continual injections for years and thus, it can be difficult to retain visual acuity in real-world clinics [8]. Furthermore, even when treatments lead to remission of the exudative choroidal neovascular membrane (CNV), subsequent visual acuities depend on the RPE that supports the physiological function of the photoreceptors. Potential alternative treatments for wet AMD patients may include surgical removal of CNV followed by the transplantation of an autologous or an allograft of RPE cells.

Clinical trials in AMD patients or animal experiments using allogeneic RPE cell grafts have previously failed, which indicates that the grafts were not able to survive because of immune attacks [9,10,11,12,13]. After transplantation, human RPE cell allografts, patches, sheets, or single cells were rejected by the host immune systems with explanted RPE cells not surviving in the retina. However, RPE cells including those established from iPS/ES cells exhibit immunosuppressive properties [14,15] with the eye also shown to be an immune privileged site [16,17]. In animal models that did use human RPE cells (xenografts), these animals displayed immune attacks, with the explanted graft ultimately not surviving [18]. In a subsequent human trial that examined AMD patients, autografts of RPE cells collected from the peripheral retina of the same patient were naturally accepted after the surgery [19]. However, there was a major issue regarding complications such as retinal detachments and proliferative vitreoretinopathy due to collecting the peripheral RPE/choroid tissues from the same patient. In 2017, we reported that an autologous RPE cell sheet prepared from iPS cells of a wet AMD patient survived without any tumors or inflammation, including any immune rejection [20]. The iPS-RPE cell sheet in this first case has remained stable and maintain the adjacent photoreceptor and choroid even after four years of observation. However, autologous iPSCs-based cell transplantation has yet to become a standard treatment because of the costs and the amount of time required. Therefore, we prepared and assessed animals using in vivo rejection models that utilized allogeneic iPS-RPE cells/sheets [1,3] in addition to preclinical studies that examined human in vitro rejection models that utilized allogeneic iPS-RPE cells established from HLA homozygote iPS cells [2]. Moreover, by using intravitreal and subconjunctival triamcinolone, we found that it was possible to control immune rejection in allografts or xenografts (human iPS-RPE cells into monkey retina) of iPS-RPE cells [3]. Therefore, the present clinical study was designed to use allografts of iPS-RPE cells from an HLA homozygous donor prepared at the Center for iPS Cell Research and Application (CiRA) iPS bank [21] for wet AMD patients with the same HLA haplotype identity. This study also used local steroid therapy for the purpose of controlling immune attacks against the grafted allogeneic RPE cells after transplantation.

## 2. Methods

### 2.1. Patients

To be eligible for the study (inclusion criteria), subjects had to be HLA-matched patients with wet AMD who had previously exhibited a limited effect after anti-VEGF injections (Table 1). Appendix A presents the full list of inclusion and exclusion criteria.

### 2.2. Study Oversight

In 2017, we announced the start of a new clinical research study (UMIN-CTR number: UMIN000026003), which included plans to use the iPS bank established at the CiRA at Kyoto University to derive RPE cells for allogeneic transplantations. In the present study, we have started to use allografts of iPS-RPE cells from an HLA homozygous donor for AMD patients who have the same HLA haplotype identity (=HLA matched).

After reviews by the Institutional Review Boards and Ethics Committees at the collaborating sites, our protocol was approved by the Certified Special Committee for Regenerative Medicine and by the Ministry of Health, Labour, and Welfare (Japan). After informed consent was obtained from all of the AMD patients, the study was conducted in accordance with the tenets of the Declaration of Helsinki. All experiments that involved the use of samples obtained from humans (and which included patients) and all animal studies were approved by the Institutional Review Board at the Kobe City Medical Center General Hospital and RIKEN BDR. The authors in this study vouch for the completeness and accuracy of the data and analyses and for the fidelity of the study to the approved protocol.

### 2.3. Primary and Secondary Outcomes

The primary objective of this study was the safety together with the feasibility of the transplantation surgery with iPS-RPE cells. Therefore, in order to achieve this objective, the primary endpoint was classified as the incidence and severity of the following: adverse events (AEs) associated with iPS-RPE cells a graft source and AEs associated with the transplantation procedure of the iPS-RPE cell suspensions. The AEs associated with iPS-RPE cells a graft source focus on: (1) Transplanted cell graft failure or immune rejection (consider the transplanted cells to have been rejected if all of the following criteria are met: a) fundoscopy does not show healthy transplanted cells; b) optical coherence tomography (OCT) shows increased retinal edema; and c) OCT shows increased subretinal fluid; and (2) excess cell proliferation or tumor formation (consider excess cell proliferation or tumor formation to have occurred if OCT shows the thickness of the transplanted cell mass to have increased by ≥200 μm compared with just after transplantation and shows increased retinal edema and subretinal fluid not associated with rejection or the recurrence of neovascularization). The AEs associated with the transplantation procedure of the iPS-RPE cell suspensions focus on (1) retinal/choroidal/vitreous hemorrhage, and (2) retinal detachment.

The secondary objectives were to evaluate the safety of the transplantation surgery with iPS-RPE cell except for the aforementioned primary endpoint, and to evaluate the efficacy for visual function. Therefore, the secondary safety endpoints included the type, incidence, and severity of AEs associated with each of the iPS-RPE cells and the transplantation procedure except for the primary endpoint, and all other AEs, which were assessed by CTCAEv4.0 JCOG.

The secondary efficacy endpoints included the following contents that evaluated at 12, 24, and 52 weeks after transplantation: OCT-based foveal retinal thickness, subretinal fluid, and retinal edema (consider the investigational treatment effective if there is a decrease), retinal sensitivity as determined by multifocal electroretinogram (ERG) and microperimetry (consider the investigational treatment effective if retinal sensitivity at the lesion site increases by 1 point or more), visual acuity (consider the investigational treatment effective if this is maintained or improved), dye leakage from CNV by fluorescence fundus angiography (consider the investigational treatment effective if dye leakage decreases), time to need for anti-VEGF treatment because of CNV recurrence and number of anti-VEGF treatment doses given during the 1-year period after transplantation (consider the investigational treatment effective if there is a decrease relative to baseline), and change in QOL (assess this according to changes in VFQ-25 scores).

### 2.4. Graft Preparation and Surgical Procedure

We prepared the grafts on the day of the surgery by suspending 2.5 × 10^5^ iPS-RPE cells into50 μL HLCR011 (Healios, Tokyo, Japan). Transplantation surgery was performed under local sub-Tenon’s anesthesia or general anesthesia. The surgical protocol included pars plana vitrectomy and the creation of a small bleb by injecting balanced saline solution into the subretinal space around the lesion followed by the subretinal transplantation of allogeneic iPS-RPE cells. The RPE cell suspension was loaded on a PolyTip^®^ cannula 25g/38g (MedOne, Sarasota, FL) with a 0.2-mL syringe installed in a custom-made fluid injector WRRK-02 (Icomes Lab, Morioka, Japan), which allowed for a gentle discharge of a precise amount of cell suspension into the subretinal space (Appendix A).

### 2.5. Genotyping of HLA Haplotype in iPS Cells and RPE Cells

The methods for the HLA genotyping in this clinical study have been previously published [22]. A peripheral blood sample was obtained from each patient, and DNA samples were obtained from lymphocyte buccal cells using a Wizard Genomic DNA Purification Kit (Promega) and stored at −20 °C. The HLA-A, -B, -C, -DRB1, -DQB1, and -DPB1 were then genotyped using a next-generation sequencing (NGS) high-resolution HLA typing method at the HLA Foundation Laboratory. The NGS HLA typing was performed based on the protocol of commercially available kits (Scisco Genetics) with Illumina MiSeq technology. The sequencing involved consecutive PCR reactions with bar codes incorporated to track individual samples followed by application to the MiSeq platform for NGS. The haplotype frequencies were calculated using the haplo.em program, evaluated by haplo.stats (version 1.6.0 provided in the public domain by the Mayo clinic http://www.mayo.edu/resea rch/docum ents/manua lhapl ostat s/doc-20167 217.) software package operated in the R language.

### 2.6. Methods of Lymphocytes-Graft Cells Immune Reaction (LGIR) Tests

Results of the LGIR tests were used to diagnose the rejection after transplantation. We first collected blood cells from the AMD patient, and then separated the peripheral blood mononuclear cells (PBMC). Subsequently we added iPS-RPE cells or allogeneic B cells as a positive control (PC), and then evaluated the immune reactions by Ki-67 FACS and IFN-γ ELISA. Both target cells were irradiated (20 Gy) before the assay. As compared to the control data (PBMC only), the patient’s lymphocytes exhibited a significant reaction against transplanted iPS-RPE cells in vitro (e.g., proliferation of T cells). This was considered to be “positive” for the immune rejection. We used the LGIR tests, as this made it possible to find which of the cells actually proliferate in PBMC after co-cultures. We monitored CD4^+^ helper T cells, CD8^+^ cytotoxic T cells, CD11b^+^ monocytes/macrophages, CD19^+^ B cells, and CD56^+^ NK cells using Ki-67 FACS analysis. In addition, we also measured IFN-γ in the supernatants of the co-cultures. We evaluated results of these six contents (proliferation of CD4, CD8, CD11b, CD19, CD56, and up-regulation of IFN-γ) in PBMC co-cultured with RPE or B cells (PC) and then compared the results with the data for PBMC only – LGIR positive = > 3 positive, LGIR suspected positive = 2 positive, LGIR negative = 1 or 0 positive. The specific details are presented in the methods [2].

In order to determine the immune attacks after transplantation, we also performed an assay for detecting RPE-specific antibody (RSA) using transplanted iPS-RPE cells and the patient’s serum before or after surgery. After immunohistochemistry (IHC), we evaluated the RPE staining by serum or antibody using confocal microscopy or by measurements of mean fluorescence intensity. We previously reported the details of these measurements in our methods [3].

### 2.7. Statistical Analysis and Data and Materials Availability

The study was designed to primarily evaluate the safety of the transplantation surgery with iPS-RPE cells in a small number of patients. Accordingly, the sample size was not determined to allow the efficacy to be evaluated with statistically sufficient precision or power, and instead a sample size of five patients was targeted. The analysis population included all the participants who had received at least part of the transplantation surgery with iPS-RPE cell. In principle, the data are shown as observed for each of these participants or summarized with the use of descriptive statistics, with no imputation of missing data. Further details are provided in the protocol and statistical analysis plan.

To protect patient privacy and comply with relevant regulations, identified data are unavailable. Requests from qualified researchers with appropriate ethics board approvals and relevant data use agreements for de-identified data will be processed by the Kobe City Medical Center General Hospital. To request access please contact the office by email: shigeki_yamasaki@kcho.jp

## 3. Results

### 3.1. Study Design

In this study, we used iPS-RPE cells from an HLA homozygous donor (six HLA gene loci: *HLA-A*, *HLA-B*, *HLA-C*, *HLA-DRB1*, *HLA-DQB1*, and *HLA-DPB1*). We initially established iPSCs using non-integrating episomal vectors [23], with these cells subsequently differentiating into RPE cells [24]. These allogeneic iPS-RPE cells were evaluated for quality and safety tests before transplantation in accordance with our protocol and our previously described methods [20]. iPS cells and iPS-RPE cells were also assessed by genome analyses such as whole genome sequencing and exome sequencing, (Appendix A), and by the expression of cancer-related genes (Figure 1). We also present the results of the residual vector test in the iPS cells (Appendix A). The Appendix A present the quality and safety assessments, and the genome analysis in the iPS cells and RPE cells. The Appendix A also presents additional details on the iPS cells and RPE cell preparation, patient recruitment, HLA typing, transplantation, and postoperative examination including immunological tests.

### 3.2. iPS Cells and RPE Cell Generation, Recipient, and Surgery

In the first part of the experiment of transplanted cells, we established RPE cells from human iPS cells (a HLA homozygous donor from the iPS bank). As seen in Figure 2A, the morphology was hexagonal and contained pigmentation. The cells exhibited phagocytosis of shed photoreceptor rod outer segments (Figure 2B), and expressed RPE-specific markers (Figure 2C). In addition, the RPE cells were found to produce large amounts of pigment epithelium-derived factor (PEDF) and VEGF in the cultures (Figure 2D). The marker and function for the iPS cells-derived RPE cells were similar to that observed with our previous RPE cells [20]. Before transplantation, we used these iPS-RPE cells to test the tumorigenicity. Immunodeficient mice (NOG mice) were used to test the tumorigenic potential of the RPE cells. No tumors were observed in these mice (Appendix A). Appendix A summarize the results of the quality control tests in these iPS-RPE cells.

In the present study, we used iPS-RPE cells (line QHJI01s04) from a HLA homozygous donor that contained the following six HLA gene loci: *A*24:02, B*52:01, C*12:02, DRB1*15:02, DQB1*06:01,* and *DPB1*09:01* (Figure 2E). The established iPS-RPE cells constitutively expressed HLA-class I (A, B, C) and did not express class II (DR, DQ, DP: Figure 2F and Appendix A). iPS-RPE cells exposed to recombinant IFN-γ expressed both HLA molecules, similar to that shown in our previous report [2].

A typical subject in the present clinical study was a wet AMD patient with an atrophic RPE lesion over the CNV with periodic recurrences while undergoing anti-VEGF treatment, and whose neural retina was at risk of degeneration (Figure 2G). As seen in Table 1, a total of five male Japanese patients with wet AMD were enrolled in this study (*n* = 5, age 64–80 years), and all patients were HLA haplotype identity that matched the above HLA homozygote iPS-RPE cells (Table 2). In fact, when we examined the HLA genotyping for our AMD patients (*n* = 105, mean age, 76.0 ± 7.8 years) in our hospital [22], there was an 18.8% HLA haplotype identity that matched the six HLA gene loci (*A*24:02-B*52:01-C*12:02-DRB1*15:02-DQB1*06:01-DPB1*09:01*) in the AMD patients. After informed consent was obtained for the present patients, we enrolled the wet AMD patients who had the same HLA haplotype identity in this study.

The first surgery case was performed on March 28, 2017. The four institutes that took part in this clinical study included, RIKEN, CiRA, Kobe City Medical Center General Hospital, and Osaka University in Japan. After initially injecting allogeneic iPS-RPE cell suspensions (~2.5 × 10^5^ cells, single cell suspensions) in the subretinal space, this was followed by a one-shot intravitreous injection of anti-VEGF drug (intravitreous injection of aflibercept: IVA) and local steroid (intravitreous injection of triamcinolone acetonide: IVTA) (Figure 2H). All the surgeries were performed as planned and completed without any complications. Appendix A summarize the results of the immunological tests.

### 3.3. Primary Endpoint

The primary endpoint of this study was the safety of the investigational treatment, i.e., the presence of any adverse events (AEs) attributable to iPS-RPE transplantation therapy as listed in our protocol. All the registered patients completed a 52-week follow-up after transplantation, and during this period, we observed no case with AEs associated with either 1) iPS-RPE cells such as poor engraftment of transplanted cells, immune rejection, excessive cell proliferation, and tumorigenicity (0/5 cases: 0.0%), or 2) surgical procedure such as retinal/choroidal/vitreous hemorrhage or retinal detachment (0/5 cases: 0.0%).

All the other adverse events are listed in Table 3. We observed mild inflammation and suspected mild rejection in two patients (Case 1 and 2); these were well controlled and were included in *other* AEs because these events were not conclusively attributed to the presence of graft cells. We had one case with retinal edema associated with epiretinal membrane (ERM) ((Case 2, listed as a serious AE by iPS-RPE cells and by transplantation surgery in Table 3), which was resolved by surgical removal of ERM with no complication. Corneal epithelial detachment was observed in 1 eye (Case 1) and one case had aseptic endophthalmitis because of intraocular triamcinolone injection as a part of transplantation procedure (Case 5). Details of these AEs are also described below in *overall clinical courses* and *immunological laboratory tests*. None of these AEs resulted in discontinuation of our study.

### 3.4. Secondary Endpoints

At 52 weeks after transplantation, we evaluated the efficacy as following: OCT-based foveal retinal thickness, subretinal fluid, and retinal edema, retinal sensitivity (multifocal ERG, microperimetry), visual acuity (BCVA), dye leakage from CNV by FA, time to need for anti-VEGF treatment, and change in QOL (VFQ-25 scores). Appendix A shows the secondary endpoint of each patient comparing with pre-operation and 52 weeks. Although RPE graft cells were not placed substantially enough in the macular lesion in most cases to reasonably affect the clinical courses of the background disease, the results of each item in the secondary endpoint as efficacy evaluation are individually listed in Table 1 and Appendix A.

### 3.5. Overall Clinical Courses

Table 1 shows the patient characteristics and the clinical features of each patient during the 1-year follow-up period. Although the macular lesion immediately became dry after the surgery with IVA/IVTA in all five cases, four cases except Case 5 had recurrences of the intra- or subretinal fluids during the 1-year follow-up period. These four patients agreed to undergo additional anti-VEGF therapies upon recurrence, with the number of additional injections during the 1-year follow-up shown in Table 1. The survival of the transplanted RPE grafts was confirmed by the presence of an area with increased pigmentation that occurred over time after the transplantation. This area became evident by 6 months in all five of the cases. While there was a substantial pigmented area present inside the macular arcade in two cases (Cases 1 and 2 in Figure 3), there were only small patches of the pigmented area present inside the arcade in three cases (Cases 3–5, Appendix A). This was most likely due to the insufficient control of the cell delivery to the targeted area during the surgery or due to an unexpected dispersion of the graft cells before settling down after the surgery. In all cases, the pigmented area was stable for the remainder of the observation period regardless of the presence of the recurrence of AMD, or the observation of active polyps in PCV networks at irrelevant positions from the graft site at the 1 year visit in 3 eyes (Cases 1, 3, and 4 in Figure 3 and Appendix A).

ERM developed in all cases. However, even though the eye in Case 2 developed diffuse edema that did not respond to either IVA or IVTA at 6 months, there was no worsening of the BCVA in this patient. There was rapid resolution of the edema after the ERM removal surgery at 7 months after the transplantation (Appendix A). Although RT-PCR showed that the removed ERM contained pigmented cells and was positive for RPE markers, the tissues did not contain any inflammatory cells/factors (Appendix A). This suggests that the ERM was of graft cell origin and that the immune rejection may not have been specifically relevant in the outcome.

In the same eye of Case 2, the subretinal iPS-RPE cells were also observed to be stable, which was quantitatively assessed as a decrease in the size of the window defect (WD) that was evaluated on the binary images constructed from the early phase fluorescein angiography (FA) (Figure 4). First, the FAG images at the pretreatment (Figure 4A) and that at 1 year after treatment (Figure 4B) were processed by the binary imaging (Figure 4C; left panel) and removing vessel images manually (Figure 4C; right panel). The WD area of the FAG image at those two different clinical time points were quantitated respectively, and resulted that the WD area at 1 year after treatment (Figure 4D) was decreased compared to that at the pretreatment (Figure 4E). The polarization-sensitive OCT (PS-OCT) images that appeared to show melanin containing cells as high-entropy signals were reconstructed into the RPE en face entropy map (Figure 4F). This map shows where the mean entropy of the 3-mm-diameter central circle increased over time, and where the low-entropy area (<0.2 below the normal range) in the central circle decreased over time after the transplantation (Figure 4G and Appendix A). The high-entropy line on the surface of CNV tissue became evident at the graft site at 12 months after transplantation (Figure 4H). These observations suggested that the transplanted RPE cells survived and led to the reconstructed RPE layer of the atrophic lesions in the macula region.

As to the immune responses, which was a major concern for this clinical trial, the mild clinical sign of rejection that was observed in only one case consisted of a subtle change in the subretinal fluid over the graft cells at 5 weeks after the transplantation. However, FA indicated that there was no evidence of leakage at 1 month (Case 1, Table 1 and Figure 5).

### 3.6. Immunological Laboratory Tests

Since the recurrence of AMD and graft rejection often share very similar clinical features, laboratory monitoring provided important clues in the detection of immune rejections in this clinical study. The laboratory tests that were performed included LGIR tests using graft RPE cells and blood cells, peripheral blood mononuclear cells (PBMC), and RPE-specific antibody (RSA) tests that were used to detect alloreactive antibody in the serum. The specific details for the methods for both examinations have been previously reported [2,3]. The results of the LGIR and RSA tests are summarized in Appendix A, respectively, with the latter tests negative in all cases, while three cases exhibited some positive LGIR test results with or without a clinical sign (Cases 1, 2, and 5).

In the LGIR tests, CD4^+^/Ki-67^+^ (proliferative helper T cells) and CD11b^+^/Ki-67^+^ (proliferative monocytes) were evaluated using flow cytometry. As seen in Figure 5A, Case 1 showed an obvious immune-response. As compared with PBMC alone (No RPE), there was a great increase in the number of proliferative CD4^+^ T cells and CD11b^+^ monocytes in PBMC at 4 weeks after being exposed to graft-equivalent iPS-RPE cells (PBMC plus RPE). Although color fundus (Figure 5B) and autofluorescence fundus (Figure 5C) did not find any inflammatory signs, there was a subtle subretinal fluid accumulation over the grafted RPE cells found by OCT at 5 weeks after transplantation (Figure 5D). These results indicated that the LGIR test was positive prior to the appearance of the clinical signs of ocular immune attack. At 8 weeks, the LGIR tests with the CD4^+^ T cells were almost completely negative against the transplanted iPS-RPE cells (Figure 5E), and color fundus (Figure 5F) and autofluorescence fundus (Figure 5G) did not find any abnormal signs. Moreover, OCT showed that the subretinal fluid disappeared after the first sub-Tenon conjunctival injection of TA (STTA) (Figure 5H).

In addition, FA examination did not find any inflammatory signs such as graft leakages after surgery (Figure 5I). CD11b^+^ cells, however, were still proliferative against iPS-RPE cells, which finally decreased to the pre-operative baseline at 36 weeks after the three STTA injections (at 6, 12, and 21 weeks) (Figure 5J). In addition, despite these immune reactions, the normal photoreceptors over the graft iPS-RPE were apparently not damaged (Appendix A).

Appendix A shows the LGIR results for Cases 2–5. In Case 2, although there was an increase in the CD4^+^/Ki-67^+^ double-positive cells at 12 weeks after surgery, we did not find any ocular inflammatory signs including at the graft site. After the STTA, these double-positive cells gradually decreased. In Cases 3 and 4, we observed no inflammatory signs in the eye, and the LGIR tests were also negative during the 1-year evaluation (Appendix A).

In Case 5, the patient had triamcinolone-related sterile endophthalmitis soon after the surgery, starting around day 1 (Appendix A). At that time, all of the LGIR tests were negative. At 24 weeks, although an examination showed that there was an increase in the CD11b^+^/Ki-67^+^ double-positive cells (monocytes) (Appendix A), the patient was not treated and only observed, as we found no rejection signs in the eye. The double-positive cells eventually decreased without any further treatment.

Appendix A list the concomitant drugs and the therapies of each patient during the 1-year follow-up period. No abnormal findings from blood tests in the five cases were observed at 4, 12, 24, and 52 weeks after transplantation. Appendix A summarizes the list of all adverse events in detail, and Appendix A describes inflammatory factors for primers and probes in qRT-PCR.

## 4. Discussion

Recently, several clinical studies using RPE cell suspensions or sheets prepared from embryonic stem (ES) cells that have been conducted in AMD patients have proven the safety of these treatments [25,26,27,28,29,30]. In these studies, the protocols included systemic immune-suppression based on the results of a previous study that found allogeneic fetus RPE had been rejected when transplanted in AMD [12]. However, it was also found that some of these patients had adverse events due to systemic immunosuppressive medications [28], which can be a physical burden in elderly patients. In the present clinical study, we used iPS-RPE cell allografts that were prepared from banked iPS cells in an HLA-matched design. The strategy for the present clinical trial was as follows: (1) Determine the HLA haplotype identity in the active exudate of AMD patients, (2) use HLA homozygote iPS-RPE cells from the iPS bank at CiRA, and (3) perform simultaneous one-shot anti-VEGF and local steroid therapies during the surgical procedure. Compared to autografts, the advantages of using allogeneic iPS cells from the cell bank were considered to be a reduction in the preparation cost and time, the stability of the graft cell preparation with a uniform protocol, and the application of the graft cells for a large number of patients. In fact, we transplanted graft iPS-RPE cells quickly and intentionally in five AMD recipients in this study, whereas it took 10 months to prepare the iPS-RPE cells in a previous study with autografts [20].

In Japan, we can cover around 80% of Japanese patients if we can find 70 donors of the HLA-3 locus homozygote (HLA-A, B, and DRB1) [21]. In fact, when we examined the HLA genotyping for our AMD patients (*n* = 105, mean age, 76.0 ± 7.8 years) in our hospital, there was an 18.8% HLA haplotype identity that matched the six HLA gene loci (*A*24:02-B*52:01-C*12:02-DRB1*15:02-DQB1*06:01-DPB1*09:01*) in the AMD patients [22]. After informed consent was obtained for the present patients, we enrolled the wet AMD patients who had the same HLA haplotype identity in this study. In our previous preclinical studies [1,2,3], we demonstrated that there was no rejection between recipients with HLA haplotype identity and HLA homozygote iPS-RPE cells. Our results showed that human HLA homozygote iPS-RPE cells failed to be recognized by allogeneic, but HLA-matched blood cells (i.e., lymphocytes in the LGIR tests) in vitro, whereas these RPE cells were recognized and stimulated lymphocytes under HLA-mismatched combinations [2]. Furthermore, there was a good survival for the allografts of monkey iPS-RPE cells/sheets and only a mild inflammation in the retina when MHC homozygote iPS-RPE cells were transplanted to monkey models of the MHC haplotype identity [1]. In these monkey models, MHC-matched transplantation with RPE allografts did not have T cell and microglia/macrophages invasions in the retina, and thus, the grafts survived with the use of only local steroids such as IVTA and STTA treatments [3].

Based on these preclinical studies, the major aim of the present study was to see if we could achieve successful iPS-RPE transplantation without having to use a systemic immune suppressant for the wet AMD patients whose MHC (HLA) matched with the donor iPS cells (HLA-class I and class II matched). Moreover, the eye is also known to be a unique organ with a unique environment that has both anti-inflammatory and immunosuppressive mechanisms [16,17]. Thus, because of this, we decided to administer local steroid-based medications (e.g., IVTA/STTA) to suppress inflammation after vitreous surgery. Steroids suppress activation of T cells such as inflammatory cytokine production, IL-2, IFN-γ, and TNF-α. Furthermore, this drug also suppresses the activation of microglia (e.g., proliferation) [31]. In the present clinical study, although additional local steroid injections (STTA) were given in two cases upon positive LGIR test findings, the pigmented cells observed after transplantation were found to be stable with increasing pigmentation over time in all five of the cases. Although systemic administration of immunosuppressive drugs still needs to be considered when a patient develops severe ocular inflammation, the present results suggest that these systemic drugs might not be needed if the HLA 6 loci are matched during the transplantation.

In the present study, we performed two immunological examinations using the patient’s blood: (1) LGIR tests in PBMC, and (2) detection of RPE-specific antibody (RSA tests) in the serum. The recurrence of AMD and immune rejections often share the same ocular findings, such as subretinal fluids and retinal edema. In this clinical trial, we determined that the LGIR test was a key for diagnosing immune rejections, and thus, when we subsequently found the LGIR tests to be positive in Case 1, treatment of this patient by local steroid injection proved to be immediately effective. More importantly, the LGIR tests at 4 weeks were positive prior to the ocular immune rejection signs in which OCT showed there was a slight accumulation of subretinal fluid over the grafted RPE cells at 5 weeks (see Figure 5). Although in all cases the humoral immunity test was negative (=no detection of RSA in the serum), the cell-mediated immunity (=LGIR positive in PBMC) was positive in one case. In addition, we found two LGIR tests that were suspected to be positive in PBMC from Cases 2 and 5, although we did not find any inflammatory signs in these eyes during the follow-up. Thus, at the present time, we cannot yet guarantee the high specificity and sensitivity of these examinations for immune rejections. In order to definitively prove the applicability of these, we will need to have larger patient cohorts in our next clinical study. Moreover, we have as of yet no definitive answer as to how the immune rejection in the eye occurred in only one case, even though several events such as iPS-associated immune attacks [32], minor antigens, natural killer cell attacks (e.g., KIR ligand mismatched) [33], and exposure to auto-retinal antigens have been reported in other cases. We are now in the process of investigating the mechanisms of RPE-related immune rejections after transplantation.

The present clinical trial also uncovered a couple of other issues that will require further elucidation in the future. One of these issues involves the dispersion and backflow of graft cells into the vitreous space, which was considered to be a cause of epiretinal membrane (ERM) in all cases. Although ERM is generally not a serious complication and is indeed removable when it causes complications such as persistent edema, it is still better that patients not to have to undergo any extra surgeries. Therefore, we need to optimize the surgical method that is used to deliver graft cells in the targeted area with minimal dispersion. Another issue involves the elevation of the intraocular pressure (IOP), especially in a steroid responder. Cases 2, 3, and 4 all exhibited an elevated IOP that was eventually controlled with glaucoma medication and/or by changing the use of the steroid eye drops.

## 5. Conclusions

We confirmed the safety of the HLA-matched allogeneic iPS cells-derived retinal cell transplantation for a period of at least one year. We were able to control the immune attacks using HLA homozygote iPS-RPE cells and local steroids in transplanted wet AMD patients. In all cases, RPE grafts survived during a 1-year observation period. LGIR tests using patient’s blood cells and transplanted RPE cells proved to be useful for the rapid diagnosis of immune attacks. The recurrences of the polyp lesions during the 1-year follow-up period were observed after transplantation in many of these patients. In this study, the grafted cell location was not sufficiently controlled to access the efficacy of this therapeutic approach. We need to further optimize the surgical procedure for controlled cell delivery to evaluate the suitable cases for the human iPS-RPE cells transplantation in the future studies.

## Figures and Tables

**Figure 1 jcm-09-02217-f001:**
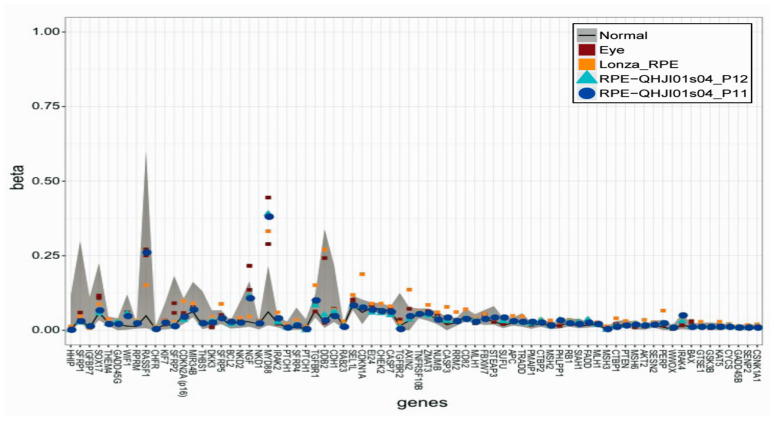
Beta values of the genomic blocks representing transcription start sites of cancer related genes. The beta values of RPE cells differentiated from iPS cells (QHJI01s04 iPS-RPE cells, *n* = 2) are comparable to those calculated for the eyes and control RPE cells (human primary RPE cells: Lonza RPE). The regions of the min-max beta values of the normal cells are shown in the gray color.

**Figure 2 jcm-09-02217-f002:**
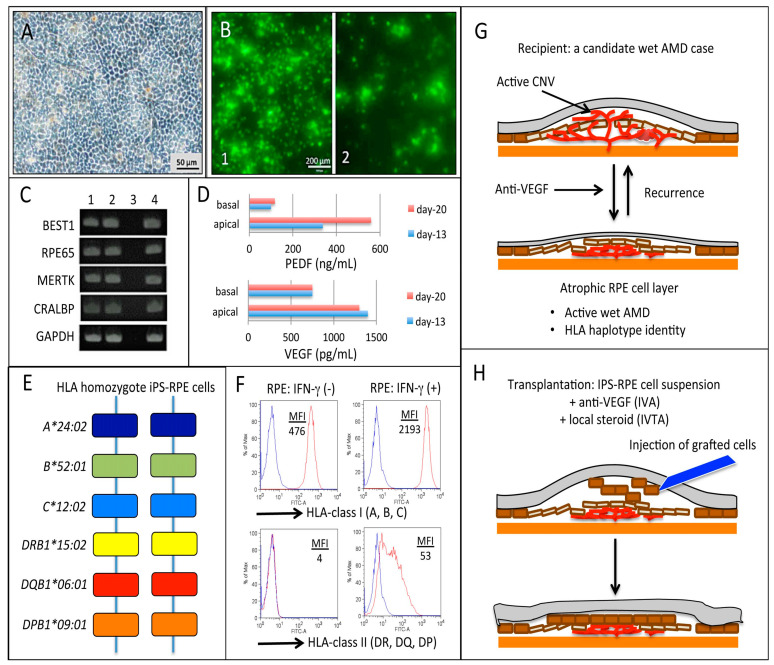
Morphology, quality control tests, and expression of HLA molecules in established RPE cells from HLA homozygous iPS cells, and a candidate wet AMD patient, and procedures of the RPE cells transplantation. This study determined the morphology (**A**), phagocytosis of shed photoreceptor rod outer segments (ROS) (**B**), expression of RPE-specific markers (**C**), and production of pigment epithelium-derived factor (PEDF) and vascular endothelial growth factor (VEGF) in the culture supernatants (**D**) from the iPS-RPE cells. (A) In our established RPE cells (QHJI01s04 iPS-RPE cells) we identified the hexagonal morphology that contained the pigment. (B) These RPE cells have phagocytic function (1. 37 °C incubation, 8 hr; 2. 4 °C incubation (control), 8 hr). (**C**) These RPE cells clearly expressed RPE-specific markers such as *BEST1* (bestrophin-1), *RPE65* (retinal pigment epithelium-specific 65 kDa protein), *MERTK* (Mer tyrosine kinase), and *CRALBP* (cellular retinaldehyde binding protein). GAPDH was the internal control in the cells. 1. QHJI01s04 iPS-RPE cells, line No. 1, QHJI01s04 iPS-RPE cells, line No. 2, 3. Water (negative control in PCR), 4. Human RPE cells (positive control in PCR). (**D**) The culture supernatants from QHJI01s04 iPS-RPE cells contained PEDF and VEGF: PEDF-apical (*n* = 3, Day-13), 336.95 ± 48.85 ng/mL, PEDF-basal (*n* = 3, Day-13), 99.25 ± 3.45 ng/mL, VEGF-apical (*n* = 3, Day-13), 1400 ± 0 pg/mL, VEGF-basal (*n* = 3, Day-13), 750 ± 50 pg/mL. We obtained similar data for the Day-20 supernatants. (**E**) In the present study, we used iPS-RPE cells (line QHJI01s04) from the HLA homozygous donor, HLA gene loci: *A*24:02, B*52:01, C*12:02, DRB1*15:02, DQB1*06:01,* and *DPB1*09:01*. (**F**) Expression of HLA-class I (A, B, C) and class II (DR, DQ, DP) on iPS-RPE cells. Right panels indicate IFN-γ pre-treatment. MFI—mean fluorescence intensity. We also determined the transplantation strategy for a candidate wet AMD patient (recipient: **G**) and the procedures for the transplantation (**H**). We transplanted RPE cell suspension (single cells: no sheet) into the subretinal space of the patients at this time.

**Figure 3 jcm-09-02217-f003:**
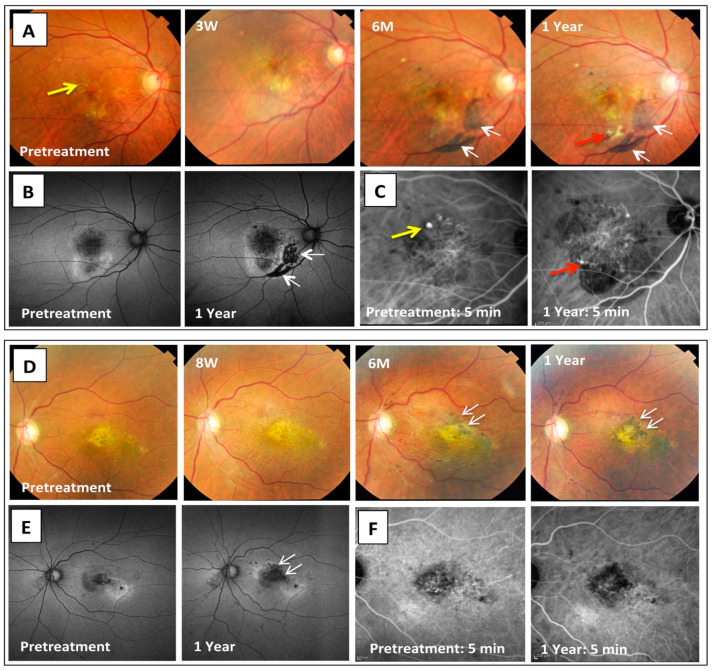
Color fundus, auto-fluorescence (AF), and indocyanine green angiography (IA) images before and at 1 year after the iPS cells-derived RPE cell transplantation in two cases. Representative fundus images during the 1-year follow-up are shown for Case 1 (**A**) and Case 2 (**D**). After surgery, grafted RPE cells were seen in the subretinal space around 3 and 8 weeks, respectively. Pigmented cells became visible by 6 months after the transplantation with a stable pigmented area observed at 1 year in both cases (white arrows). AF images (**B**,**E**) show a dark area that corresponds to the grafted cells at 1 year after the transplantation (white arrows). In IA images of Case 1 (**C**), yellow arrows indicate the presence of polyp lesions before the treatment. After 1 year, Case 1 developed a new polyp that was associated with hemorrhagic lesions on the lower margins of the PCV network (red arrows). On the other hand, there were no changes in the findings of the IA examination in Case 2 as compared with the pretreatment (**F**).

**Figure 4 jcm-09-02217-f004:**
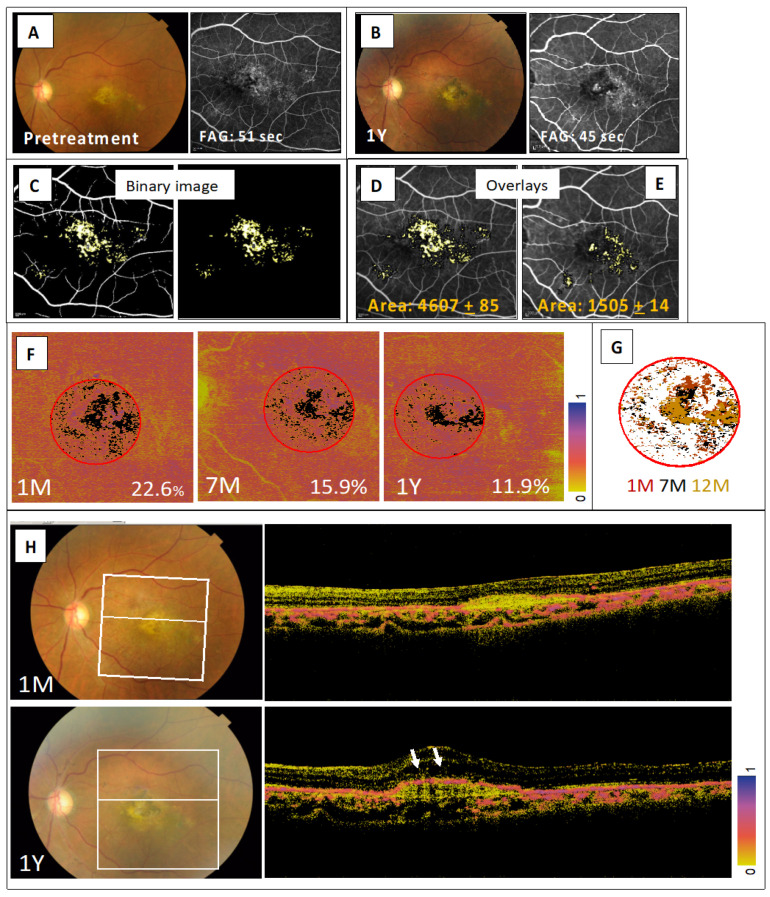
Quantitative assessment of transplanted iPS-RPE cells in Case 2. Color fundus photographs and early phase fluorescein angiography (FAG) images obtained at pretreatment (**A**) and at 1 year after treatment (**B**). (**C**) Window defect (WD) shown by binary image processing of the FAG images at pretreatment. The FAG image was processed by binary imaging (C; left panel) and, then the vessel images was removed manually (C; right panel). Overlays of WD binary image for the pretreatment FA (**D**), and the FA at 1 year after treatment (**E**). WD area is displayed in pixels. (**F**) PS-OCT evaluation showed that the presence of pigmented RPE cells indicated the high-entropy area was above the RPE basement membrane and that there was a time-dependent decrease in the low-entropy area seen within the fovea-centered 3-mm-diameter circle. Low-entropy area at 1, 7, and 12 months after transplantation is shown in black on the entropy map for each time point with the percentage of black area within red circle. (**G**) Low-entropy areas for each of the time points were aligned with the retinal vessels and are shown on the right panel to demonstrate the consistency in the pattern. (**H**) Time course of a representative section view at the white line on the left color fundus image, which shows the continuity of the presumably melanin-containing RPE cells covering the surface of the fibrous tissue at 12 months after transplantation.

**Figure 5 jcm-09-02217-f005:**
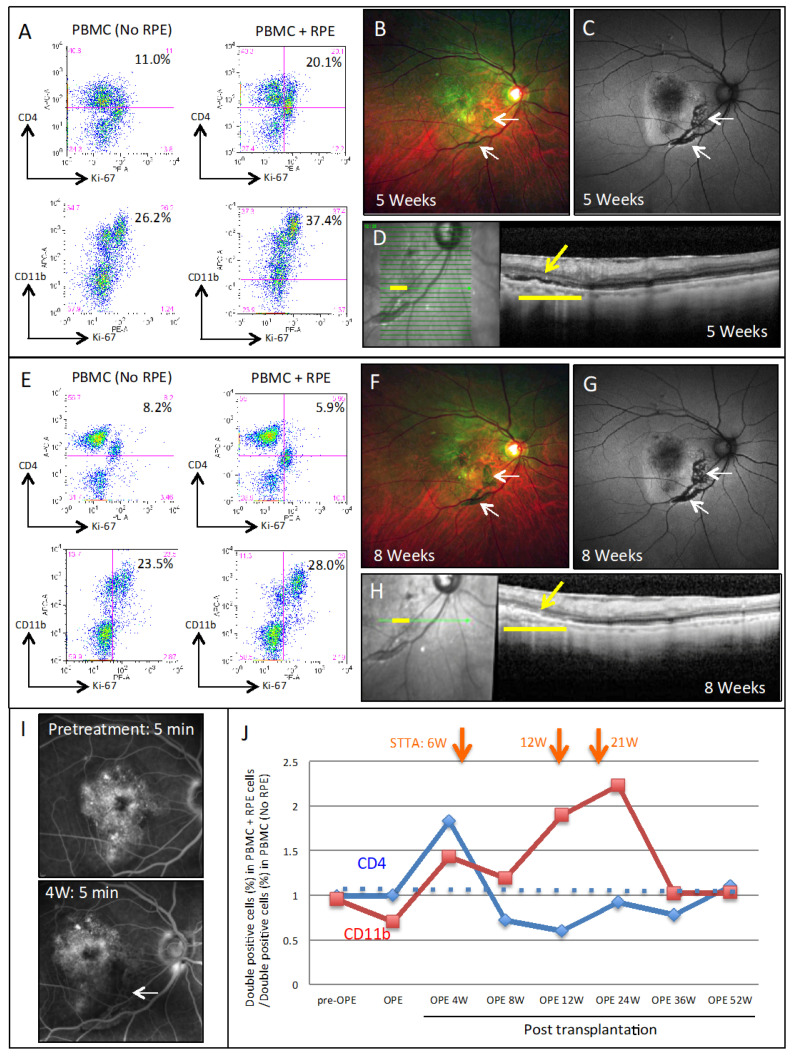
Findings of immune attacks in the eye of a transplanted AMD patient (Case 1). (**A**) FACS results in the lymphocytes-graft cells immune reaction (LGIR) test at 4 weeks after surgery in the AMD patient. For the LGIR test, there was a great proliferation of the CD4^+^/Ki-67^+^ (proliferative helper T cells) and CD11b^+^/Ki-67^+^ (proliferative monocytes) in PBMC as compared with the PBMC only (no RPE). (**B**) Color fundus photographs at 5 weeks after surgery of the transplanted AMD patient (Case 1), and (**C**) the autofluorescence (AF) fundus photographs (arrows, grafted RPE cells). (**D**) OCT image, grafted area (yellow bar) at 5 weeks. There were small amounts of subretinal fluids in the subretinal space (yellow arrow). The LGIR test at 8 weeks (**E**), showed that there was poor proliferation of the CD4^+^ T cells in PBMC that were exposed to transplanted iPS-RPE cells. On the other hand, the proliferation of the CD11b^+^ cells against RPE cells still remained. (**F**) Color fundus and AF photographs (**G**) at 8 weeks after surgery in the same patient. (**H**) OCT shows that the subretinal fluids disappeared after local steroid treatment. (**I**) Panels show fluorescence angiography images before and at 4 weeks after transplantation, (arrow, grafted RPE cells). There were no leakages from the grafted cells observed in the fluorescence angiography. (**J**) LGIR FACS results after surgery of the AMD patient (Case 1). There was a decrease in the CD4^+^/Ki-67^+^ double-positive cells observed after the sub-Tenon conjunctival injection (STTA; orange arrows), and a decreased in the CD11^+^/Ki-67^+^ double-positive cells at 36 weeks after administration of STTA for a total of three times.

**Table 1 jcm-09-02217-t001:** Patient characteristics and clinical summary before and after iPS-RPE cell transplantation.

No.	Case (HLA Type)	Sex, Age	Disease (CNV Subtype)	Operated Eye	Immunological Test: LGIR Test (Clinical Sign)	Before Transplantation		Graft Cells in Macular	After Transplantation	
						BCVA (Snellen)	Number of Anti-VEGF		BCVA (Snellen)	Number of Anti-VEGF
1	Case 1: al-so-w01	M, 64 yr	Wet AMD	Right	Positive at 4W	0.08	4/year	Present	0.06	3/year
	(Haplotype Identity)		(PCV: occult)		(Subtle SRD: 5W)	(20/250)		outside lesion	(20/320)	
2	Case 2: al-so-w02	M, 75 yr	Wet AMD	Left	Suspected at 12W	0.1	2/year	Present	0.15	3/year
	(Haplotype Identity)		(occult)		(None)	(10/100)			(20/125)	
3	Case 3: al-so-w03	M, 80 yr	Wet AMD	Right	Negative	0.1	3/year	Present	0.15	2/year
	(Haplotype Identity)		(PCV: minimally classic)		(None)	(10/100)		in a small area	(20/125)	
4	Case 4: al-so-w04	M, 68 yr	Wet AMD	Right	Negative	0.09	1/year	Present	0.1	1/year
	(Haplotype Identity)		(PCV: occult)		(None)	(10/100)		in a small area	(10/100)	
5	Case 5: osk-al-so-w01	M, 68 yr	Wet AMD	Right	Suspected at 24W	0.2	3/year	Present	0.15	0/year
	(Haplotype Identity)		(PCV: occult)		(None)	(20/100)		outside lesion	(20/125)	

Data after the transplantation were collected at 52 weeks after surgery. BCVA (best-corrected visual acuity) indicates that 0.1 is equivalent to 10/100 on a Snellen chart. Number of anti-VEGF—number indicates times per a year of intravitreous injections of aflibercept (IVA) during the last 1 year before transplantation. PCV—polypoidal choroidal vasculopathy. LGIR test—lymphocytes-graft cells immune reaction test. SRD—serous retinal detachment. AMD: age-related macular degeneration. yr—years old.

**Table 2 jcm-09-02217-t002:** Results of HLA-haplotype in iPS-RPE cells and patients with AMD.

No	Cells or Case (AMD)	HLA-A	HLA-B	HLA-C	HLA-DRB1	HLA-DQB1	HLA-DPB1
1	QHJI01s04 iPS-RPE cells	24:02/-	52:01/-	12:02/-	15:02/-	06:01/-	09:01/-
2	Case 1: al-so-w01	11:02/24:02	27:04/52:01	12:02/-	12:02/15:02	03:01/06:01	02:01/09:01
3	Case 2: al-so-w02	24:02/26:01	40:02/52:01	3:04/12:02	08:02/15:02	03:02/06:01	05:01/09:01
4	Case 3: al-so-w03	02:06/24:02	44:03/52:01	12:02/14:03	13:02/15:02	06:01:06:04	04:01/09:01
5	Case 4: al-so-w04	24:02/26:02	15:01/52:01	03:03/12:02	04:05/15:02	04:01/06:01	05:01/09:01
6	Case 5: osk-al-so-w01	24:02/-	52:01/15:07	12:02/03:03	15:02/04:03	06:01/03:02	09:01/02:01

Donor cells were HLA homozygote cells and all AMD patient recipients (Cases 1—5) exhibited the HLA haplotype identity.

**Table 3 jcm-09-02217-t003:** Lists of main adverse events in all cases.

Adverse Events (AEs)	Case
Immune rejection in the eye	Case 1
Suspected rejection (No ocular signs)	Case 2
Corneal epithelial detachment	Case 1
Epiretinal membrane (ERM)	Case 2
Cystoid macular edema or retinal edema associated with ERM	Case 2 *, 4
Macular pseudohole	Case 4
Elevated intraocular pressure	Case 2, 3, 4
Triamcinolone-related endophthalmitis	Case 5 *

More detail information for all AEs in these cases is in Appendix A. * Severe AEs.

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
