# Peer review of "HLA-Matched Allogeneic iPS Cells-Derived RPE Transplantation for Macular Degeneration"

_jcm, 2020, doi:10.3390/jcm9072217_

Round 1

Reviewer 1 Report

This is an exceptionally well designed and executed small clinical trial designed to assess the safety and the immune reactions using HLA-matched allogenic  iPS cells-derived retinal pigment epithelium (iPS-RPE) transplantation in patients with exudative age-related macular degeneration. The authors note that several clinical studies using RPE cell suspensions or sheets prepared from embryonic stem (ES) cells injected into the eyes of AMD patients have documented the safety of this approach.  However these protocols required systemic immune-suppression.  Moreover, some of these patients had adverse events due to systemic immunosuppressive medications. In the present clinical study,  the authors  used iPS-RPE cell allografts that were prepared from banked iPS cells in an HLA-matched design. The strategy for the present clinical trial was to: (1) Determine the HLA haplotype identity in the active exudate of AMD patients, (2) Use HLA homozygote iPS-RPE cells from the iPS bank at CiRA, and (3) Perform simultaneous one-shot anti-VEGF and local steroid therapies during the surgical procedure. They found that compared to autografts, the advantages of using allogeneic iPS cells from the cell bank were considered to be a reduction in the preparation cost and time, the stability of the graft cell preparation with a uniform protocol, and the application of the graft cells for a large number of patients.  This reviewer agrees with their conclusion that the cell delivery strategy must be further optimized, however, the present results suggest that it is possible to achieve stable survival and safety of iPS-RPE cell transplantation for a year.

Author Response

Response: We would like to thank for their careful review of our manuscript and their helpful comments.

Reviewer 2 Report

this work is very interesting and can become a milestone. Such research gives new hope for patients with amd . The subretinic pathway seems to be the most promising.

Author Response

Response: Thank you for your comments.

Reviewer 3 Report

The present manuscript depicts a clinical study, where transplanted HLA homozygote iPS-RPE cells established at an iPS bank in HLA-matched patients with exudative age-related macular degeneration with the coadministration of local steroids without immunosuppressive medications. The authors after assessed the safety of the procedure by monitoring the presence of adverse events for one year.

The manuscript is easy to read and understand. The experiments carried out were suitable for the aims of the manuscript.

Although, as the authors state, sample size is small, the findings of the present manuscript are very interesting for a broader community, the patient evolution monitoring during a year provides an enhanced depth of the research.

In my opinion, the findings are interesting for a broader community and deserve to be published. Even though, the paper comes with a few issues, which are addressed below:

Some symbols are missing on the manuscript:

Lines 149 – 150: 2.5 × 105 iPS-RPE cells into 50 L → 2.5 × 105 iPS-RPE cells into 50 μL

Line 174 IFN- → IFN-γ; INF appears several times on the manuscript

Some text on the images provided is not legible and weird symbols appeared instead:

Revise the superposed text on Figure 4: A, B, C, D, and G.

Revise the superposed text on Figure 5: A – G

Author Response

We would like to thank for their careful review of our manuscript and their helpful comments.

Some symbols are missing on the manuscript:

Lines 149 – 150: 2.5 × 105 iPS-RPE cells into 50 L → 2.5 × 105 iPS-RPE cells into 50 μL

Line 174 IFN- → IFN-γ; INF appears several times on the manuscript

Response: Thank you for your comments. We revised the symbols such as IFN-gamma in revised manuscript.

Some text on the images provided is not legible and weird symbols appeared instead:

Revise the superposed text on Figure 4: A, B, C, D, and G.

Revise the superposed text on Figure 5: A – G

Response:Thank you for the pointed out. Actually, we had some mis-sentences in text of Fig. 4 and 5. We therefore have revised the text.